# Application of Feature Pyramid Network and Feature Fusion Single Shot Multibox Detector for Real-Time Prostate Capsule Detection

**Shixiao Wu** [1,2,*] 🆔, **Xinghuan Wang** [3] 🆔 **and Chengcheng Guo** [1,*]

1    Electronic Information School, Wuhan University, Wuhan 430072, China
2    School of Information Engineering, Wuhan Business University, Wuhan 430056, China
3    Department of Urology, Zhongnan Hospital, Wuhan 430072, China
*    Correspondence: wushixiao@whu.edu.cn (S.W.); netccg@whu.edu.cn (C.G.)

**Abstract:** In the process of feature propagation, the low-level convolution layers of the forward feature propagation network lack semantic information, and information loss occurs when fine-grained information is transferred to higher-level convolution; therefore, multi-stage feature fusion networks are needed to solve the interaction between low-level convolution layers and high-level convolution layers. Based on a two-way feature feedback network and feature fusion mechanism, we created a new object detection network called Feature Pyramid Network (FPN)-based Feature Fusion Single Shot Multibox Detector (FFSSD). A bottom-up and top-down architecture with lateral connections enhances the detector's ability to extract features, then high-level multi-scale semantic feature maps are utilized to generate a feature pyramid network. The results show that the proposed network the mAP for prostate capsule image detection reaches 83.58%, providing real-time detection ability. The context interaction mechanism can transfer high-level semantic information to low-level convolution, and the resulting convolution after low-level and high-level fusion contains richer location and semantic information.

**Keywords:** feature pyramid network; feature fusion; prostate capsule

## 1. Introduction

Early object detection techniques did not use feature fusion technology [1–3]. Following the emergence of deep learning, more and more feature fusion techniques have been used in object detection networks, including image pyramid, feature fusion for R-CNN series, Single Shot Multibox Detector (SSD), Feature Pyramid Network (FPN) in FoveaBox, Sparse R-CNN and Task-aligned One-stage Object Detection (TOOD), Feature Fusion Single Shot Multibox Detector (FSSD), Weighted Bidirectional feature pyramid Network (BiFPN) in You Only Look Once Version 4 (YOLOv4) and EfficientDet, Path Aggregation Network (PANet) in YOLOv4, and more [4–14]. An image pyramid is composed of multiple multi-resolution and multi-scale images, such as a Gauss/Laplacian pyramid. Ziming Liu proposed IPG-Net using image pyramid on 2020 CVPR to solve the problem of feature misalignment and small target deep feature loss in object detection [15]. To an extent, an image pyramid makes the results of object detection more accurate. The disadvantage is that as the pyramid level increases, the amount calculation increases and the detection speed slows down. The feature fusion work of R-CNN series makes prediction on the last layer. Taking Faster R-CNN as an example, the algorithm relies on multi-stage training to improve the detection precision, and the precision improvement of feature fusion for Faster R-CNN is limited. Each feature layer of the SSD series is detected separately. Its disadvantage is that there is less convolution at the lower level involved in detection (only one) and the location information may not be accurate enough. The feature fusion of FPN adopts the methods of bottom-up, lateral connection, and top-down, which improves the accuracy of object

detection. Its disadvantage is that the network is more complex, with accuracy improved at the expense of speed. The feature fusion strategy of FSSD is to construct the feature pyramid through bottom-up and multi-scale fusion. FSSD fuses the features of three scales and uses bilinear interpolation and $1 \times 1$ convolution to adjust the size of the feature map and the number of channels. FSSD follows the idea that the small-scale convolution layer is not useful; thus, part of the size and number of channels are discarded during feature fusion. The above works use a single pyramid network to complete feature extraction, and the results of object detection may not be accurate. YOLOv4 and EfficientDet ultized BiFPN for feature fusion, which introduces learnable weights to learn the importance of different input features while repeatedly applying top-down and bottom-up multi-scale feature fusion. PANet is based on the proposed instance segmentation framework enhances the entire feature hierarchy by using accurate low-level positioning signals, thereby shortening the information path between the low-level and top-level features; however, experiments have shown that such complex networks may not work well with small data sets.

Fusing different networks to extract features has recently become very popular. Ibtissam Bakkouri et al. proposed a multistage fusion network for skin injury recognition [16]. The most discriminant features were extracted based on the VGG, ResNet, and DenseNet networks, with three features output from the three networks fused by the feature fusion layer. Xiaojing Qiu proposed a multi-stage feature fusion network for medical image segmentation which included a basic feature expression layer, intermediate feature fusion layer, deep feature fusion layer, and feature fusion module [17]. The middle layer and the fusion layer were used to complement the feature information of the lower layer by means of convolution, channel superposition, and corresponding element addition. In this approach, the deep fusion layer uses the supplementary information of deep and shallow features while the three-level feature representation layer completes the feature aggregation process. Lingzhi Kong used DenseNet and VGG16 for feature extraction and fusion [18]. In this approach, the input of DenseNet is the original image, and the input of VGG16 is the image processed by ResNet. Their aim was to quickly achieve COVID-19 detection using an AI approach. Their proposed feature fusion network integrates different backbone networks; however, it neglects the feedback and semantic transfer of high-level convolution to low-level convolution.

The current prostate capsule research includes the surgical level, contour recognition, thickness calculation, and volume calculation. At the surgical level, Zia et al. explored the U-shaped incision of the prostate capsule in laparoscopic prostatectomy, and Alon Z. Weizer evaluated the risk of urothelial and prostate cancers in a series of patients undergoing radical cystoprostatectomy with preserved prostate capsule [19,20]. Rania Hussein used prostate shape and generalized Hough transform to automatically recognize the contour of the prostate capsule [21]. K.G. Holder et al. studied the relationship between prostate volume, prostate capsule thickness, and gland epithelial cell density in the peripheral zone (PZ) using tissue anatomical imaging technology [22]. In this paper, we utilize FPN to complete the reconstruction of low-level convolution, then take the convolution reconstructed by fusion as the basis to perform $2\times, 4\times, 8\times, 16\times$ and $32\times$ downsampling. To complete the real-time detection of the prostate capsule, we construct a new network, FFSSD, based on the FPN and FSSD networks.

This article's innovations include the following:

(1) To reduce the impact of salt and pepper noise on the object detection network, a salt and pepper noise reduction method based on edge feature preservation is proposed. Compared with fourteen other methods, this method has the highest peak signal-to-noise ratio. The proposed image denoising methods can improve the mAP of Faster R-CNN, YOLOv4, TOOD, SSD, and FSSD.

(2) Based on FPN and FSSD, a multistage bidirectional feature fusion network called FFSSD is proposed. Compared with other algorithms such as Faster R-CNN, YOLOv4, TOOD, SSD, FSSD, OWOD, Foveabox, Sparse R-CNN, and Efficientdet, the proposed algorithm has the highest mAP on the prostate capsule detection task.

## 2. Materials and Methods

### 2.1. Dataset

The images of the prostate capsule were extracted from surgical videos. Due to the interference of gastrointestinal peristalsis and other noise, salt and pepper noise was easily introduced into the images. Using the Composed Fusion (CF) algorithm to denoise pictures with noise can ensure the smooth development of subsequent object detection work. The CF algorithm is described in Algorithm 1.

---

**Algorithm 1** CF algorithm

---

**Input:** Noisy_img $v$, noisy level $x$
**Output:** $p(R)$, $s(R)$, $R$, $denoised\_img\{R\}$
 1: Function $u$ = compose($v$, $z$)
 2: Initialize $d$ = {}, $p$ = ones(0,6), $s$ = ones(0,6), $R$ = 0, $MAX$ = 0, $D$ = {2,6}
 3: set *Denoised_img*{1} = AMF($v$,$x$)
 4: set *Denoised_img*{2} = NAMF($v$,$x$)
 5: set *Denoised_img*{3} = ANN($v$,$x$)
 6: set *Denoised_img*{4} = RUN_ADF(*Denoised_img*{1},*Denoised_img*{2},0)
 7: set *Denoised_img*{5} = RUN_ADF(*Denoised_img*{1},*Denoised_img*{3},0)
 8: set *Denoised_img*{6} = RUN_ADF(*Denoised_img*{2},*Denoised_img*{3},0)
 9: MAX = PSNR(
10: **for all** $(s_p) \in D$ **do**
11:     $p(k)$ = PSNR(*denoised_img*{k},o)
12:     $s(k)$ = COMPUTE_FOM(*denoised_img*{k},o)
13:     IF($p(k) > MAX$)
14:     {$MAX = p(k)$; $R = k$}
15: **end for**
16: **return** $p(R)$, $s(R)$, $R$, $denoised\_img\{R\}$

---

In the CF algorithm, Adaptive Median Filter (AMF), Anisotropic Diffusion Fusion (ADF), Non-local Adaptive Mean Filter (NAMF), and Artificial Neural Network (ANN) algorithms are metioned, while Peak Signal-to-Noise Ratio (PSNR) and Figure of Merit (FoM) are the criteria used to evaluate image denoising. The CF algorithm steps are as follows:

(1)   The image with noise is denoised by AMF, NAMF, and ANN.
(2)   Anisotropic diffusion fusion is used to combine the image denoising results of AMF, NAMF, and ANN in pairs.
(3)   The noise reduction results of the three algorithms and the pin-two fusion results of the three algorithms are combined, then the maximum value of the combination according to the PSNR is used to obtain the final image denoising cascade optimization results.

In Figure 1, the CF is closer to the original and performs better than individual algorithms (Tristate Filter (TF), AMF, Median Filter (MF), Adaptive Total Variation L1 Regularization (ATVL1), Decision-Based Algorithms (DBA), Noise Adaptive Fuzzy Switching Median (NAFSM), NAMF, ANN, Convolutional Neural Network (CNN)-based methods, ADF-based methods, Adaptive Centre Weight Median Filter (ACMF), and Noise Adaptive Edge-Preserving Filter (NAPEF)). The areas shown in detail after image denoising are indicated by the red rectangles [23–31].

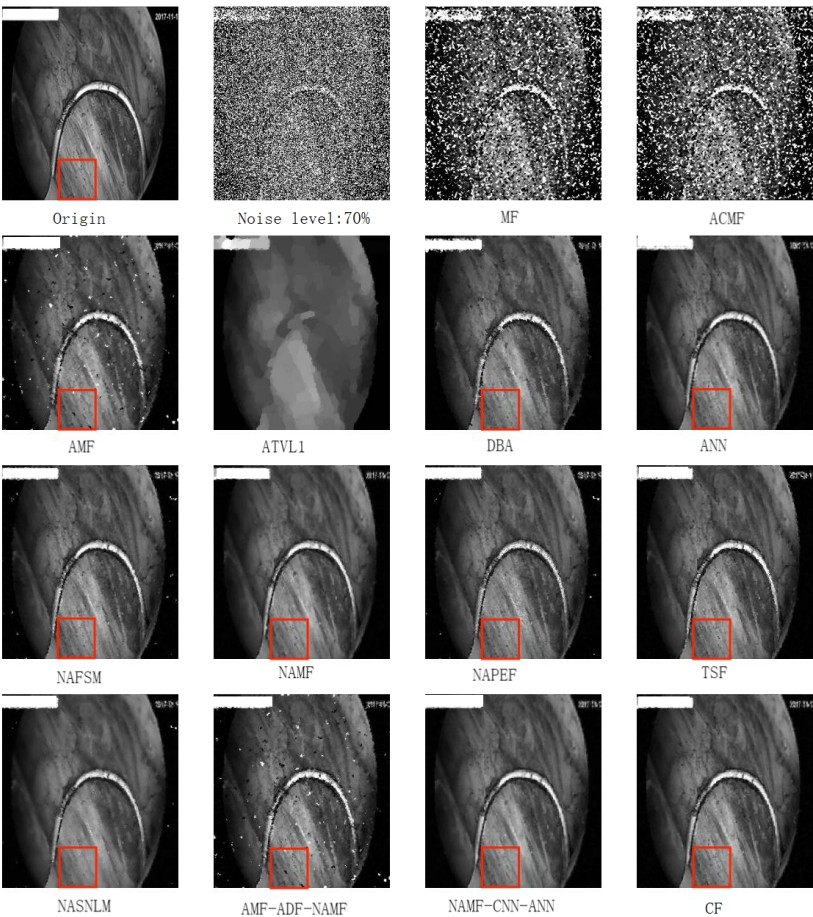

**Figure 1.** CF results for denoising, compared with fourteen different methods.

Compared to ANN and NAMF, CF performs the best, maintaining better edge detail after noise reduction; see Figure 2.

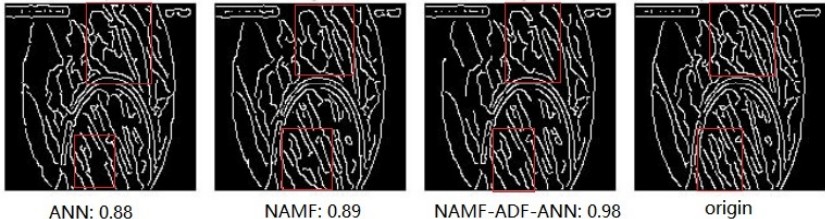

**Figure 2.** Edge detection comparision.

The proposed network was verified on three datasets, all of which contained 597 images split into 478 training and 119 test images. The datasets used consisted of the Primary Component Analysis (PCA)-based dataset, PCA+CF-based dataset, and original dataset. For the first two datasets, preprocessing was only performed on the training set, while the test set remained unchanged.

In addition, compared with other detection objects of fixed size, the prostate capsule has the characteristics of large-scale changes and different shapes. If the object detection algorithm is directly used to detect the prostate capsule from an image, the mAP of the resulting detection may not be high enough (Figure 3).

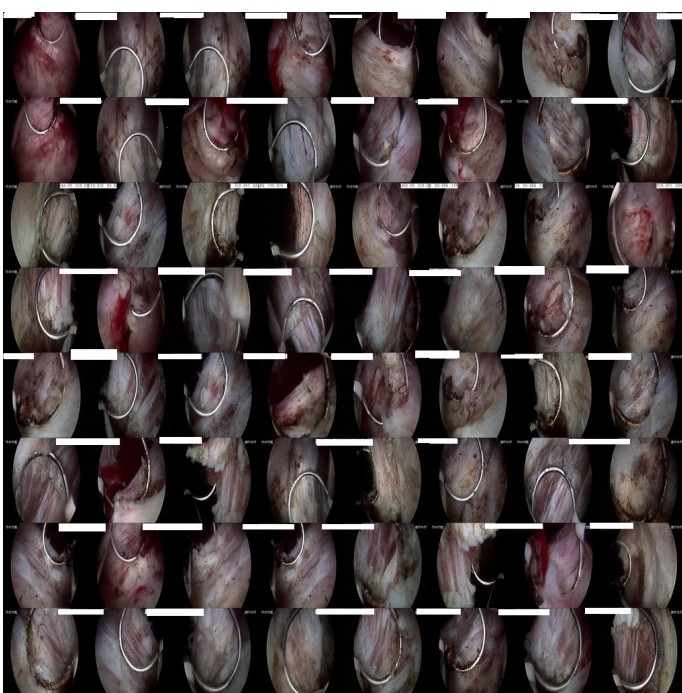

**Figure 3.** The prostate capsule has the characteristics of large-scale changes and different shapes.

## 2.2. The Proposed Network

Based on FPN and FSSD, we propose a new network called FFSSD. The feature pyramid network consists of three parts, two feature forward propagation parts and one backward propagation part. A new feature pyramid was formed by three feature propagation parts.

### 2.2.1. First Feature Forward Propagation

Feature forward propagation consists of two parts. The first feature forward propagation is based on the feature pyramid of the SSD network, and the feature propagation ends at convolution layer conv7_2 (Figure 4). Fine-grained position information is passed from the lower convolution to the higher convolution. This forward propagation belongs to the VGG16 feature extraction backbone network.

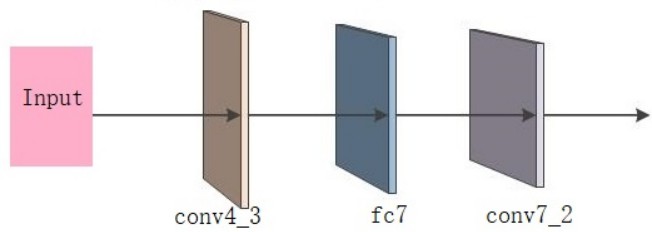

**Figure 4.** First feature forward propagation.

### 2.2.2. Reverse Feature Propagation

After the first forward feature is passed to conv7_2, reverse feature propagation begins. After the first forward feature information completes forward propagation of position information, reverse feature propagation can transmit the semantic information of the prostate capsule to the lower convolution layer. In this way, the higher-level convolution has the fine-grained information transmitted by the lower-level convolution, and the lower-level convolution has the semantic information returned by the higher-level convolution (Figure 5).

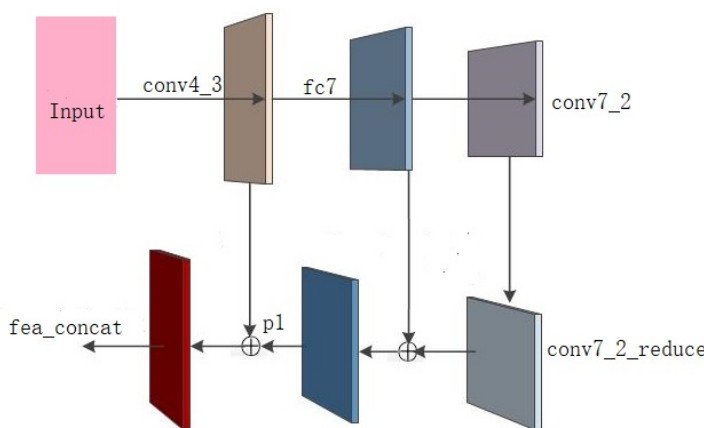

**Figure 5.** Reverse feature propagation.

As shown in Figure 5, we adjusted the conv7_2 through $1 \times 1$ convolution layers to obtain conv7_2_red uce; p1 was formed by conv7_2_reduce (2x upsampling) and fc7, while fea_concat was formed by conv4_3 and p1 ($2\times$ upsampling).

### 2.2.3. New Feature Pyramid Network

After completion of the reverse propagation, the new feature pyramid is generated by the second feature forward propagation. The second feature forward propagation builds on the first forward propagation and reverse feature propagation. The forward and reverse feature propagation of conv4_3, fc7, conv7_2 constitute an FPN feature pyramid network. Bidirectional passing (forward propagation and reverse propagation) constitutes the context interaction mechanism, which integrates and supplements the low-level location information and high-level semantic information. The three convolution layers are fused by the FPN feature pyramid to form a new convolution layer. On this basis, second feature forward propagation is carried out to generate a new feature pyramid network (Figures 6 and 7).

On the basis of the new fea_concat layer generated by FPN fusion, downsampling of $2\times, 4\times, 8\times, 16\times,$ and $32\times$ was implemented to carry out the second forward propagation of features. Compared with SSD, which is only downsampled on a single convolution layer, conv4_3, the details of the shallow convolution layer of the multistage feature fusion method are richer thanks to bidirectional feature fusion, which enables the high level to learn more information about the prostate capsule. In Figure 7, the orange dotted line box represents the FPN feature fusion network and the blue dotted box indicates the feature fusion network of the FSSD. The small dotted line box in black represents the fusion process of the lower-layer feature graphs by one transverse connection ($1 \times 1$conv) and $2\times$ upsampling with higher convolution. BatchNorm accelerates the convergence process of the convolutional network, and the $1 \times 1$conv eliminates the effect caused by upsampling. The fusion details are shown in the large dotted black box.

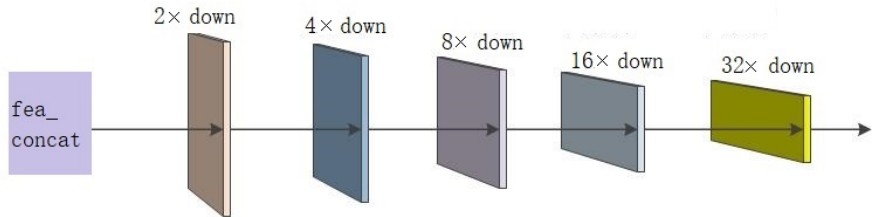

**Figure 6.** Reverse feature transfer.

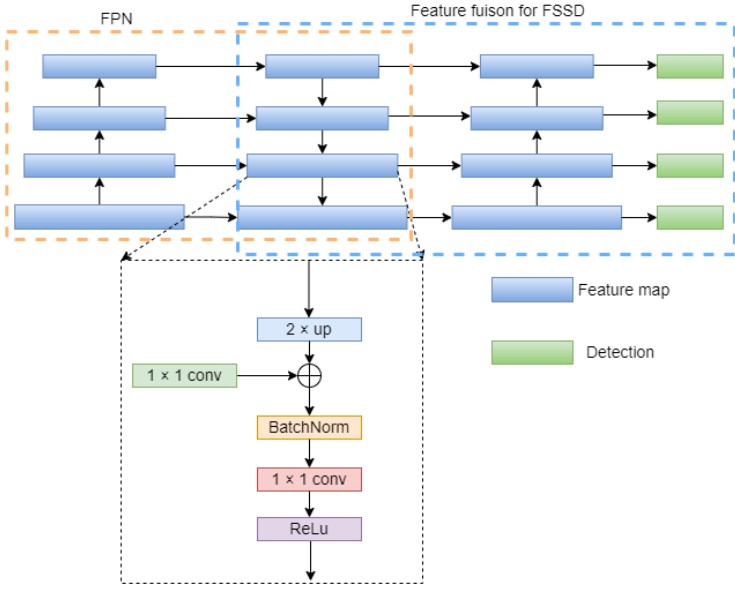

**Figure 7.** Multi-level feature pyramid network in FFSSD.

### 2.2.4. FFSSD Network

Based on the new feature pyramid network, the FFSSD network is designed as shown Figure 8.

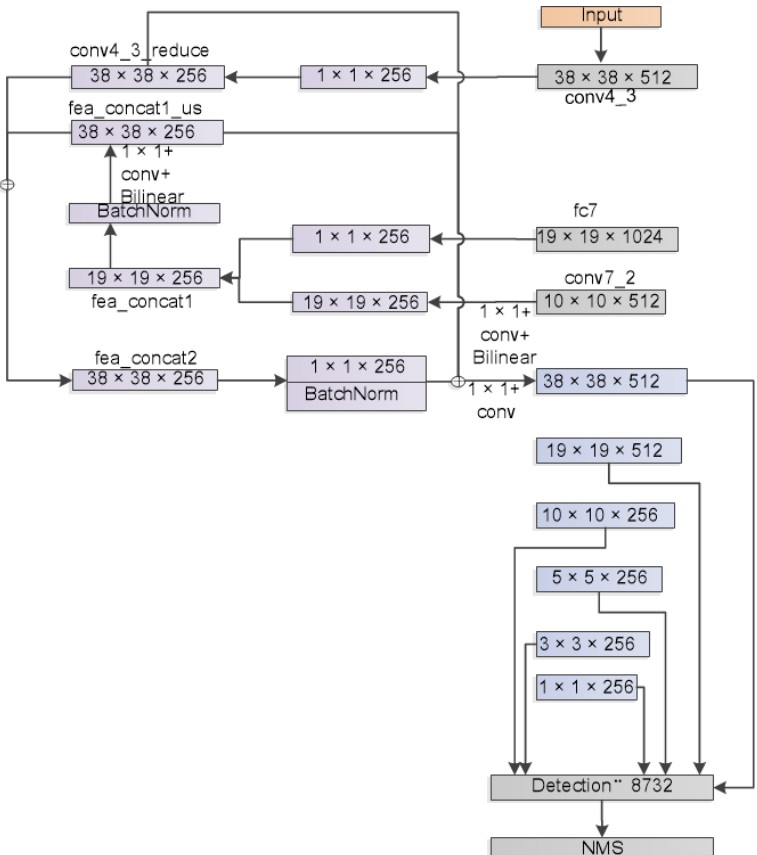

**Figure 8.** FFSSD Network.

The FFSSD backbone network is the VGG16 network. The feature fusion network consists of one FPN network and one FSSD fusion module. The FPN network consists of

conv4_3, fc7, and conv7_2 through feature forward propagation and reverse propagation. The first feature forward propagation transmits the position information of the model from the bottom layer to the top layer; the position information gradually decreases while the semantic information gradually increases. Through this initialization, the network ensures the preliminary determination of the category of the object by high-level convolution. In the second reverse propagation, the semantic (type) information is transferred from the upper level (conv7_2) to the lower level (conv4_3) and the semantic information in the lower level convolution is increased, that is, the lower-level convolution (conv4_3) now carries the location information of the prostatic envelope and has rough knowledge of its semantic information. After the fusion of semantic and positional information through bi-directional transmission of the FPN feature pyramid network, the conv4_3 fusion of positional and semantic information carries the dual information for the third feature propagation, that is, the second feature forward propagation. This time, based on fusion convolution, the fusion information is transmitted from the lower level to the upper level by bidirectional transmission, and the final category determination and location calibration are carried out.

The VGG16 backbone network of FFSSD is the same as that of FSSD. Unlike SSD, the convolution of feature maps less than $10 \times 10$ is abandoned. An FPN network consists of two feature fusion modules. There are three convolution layers involved in FPN feature fusion, namely, conv4_3, fc7, and conv7_2; the dimensions of the three convolutional layers are $38 \times 38 \times 512$, $19 \times 19 \times 1024$, and $10 \times 10 \times 512$, respectively. The convolution layer with the smallest scale is fused to form the first module through the second smallest convolution layer with twice the upsampling and adjusted channels, and the second module is fused with the feature degree of the maximum scale with twice the upsampling and adjusted channels. The two feature fusion modules of FPN were fused with conv4_3 through double upsampling and transverse connection, which together constitute the feature fusion module of FSSD.

The feature extraction layer involved in detection in the network structure is based on the result of two-stage fusion, and the result layer of two-stage fusion is obtained by $2\times$, $4\times$, $8\times$, $16\times$, and $32\times$ downsampling.

## 3. Results

The framework used for training the deep learning networks was Caffe/pytorch. The hardware environment for network training was an Intel Core-i7-8700 (Microsoft Corporation, Redmond, WA, USA) , the main frequency of the CPU was 3.2 GHz , the running memory was 8 GB , the graphics card model was an NVIDIA GTX 1070/Tesla v100 (Microsoft Corporation, Redmond, WA, USA) , and the operating system was Linux Ubuntu 64-bit (Red Hat, Raleigh, NC, USA) . The learning rate of the first 500 iterations was 0.0001, and the batch_size used for training was 1.

### 3.1. Criteria for Evaluation

In object detection techniques, algorithms are generally evaluated on their mAP (mean Average Precision) and FPS (Frame Per Second). The mAP indicates the Average Precision *AP* of all detection classes, while the *AP* refers to the area enclosed under the Precison–Recall (PR) curve, with a larger area being better. Its formula is defined as follows:

$$Precision = \frac{TP}{TP + FP}$$
$$Recall = \frac{TP}{TP + FN} \tag{1}$$

*Precision* generally refers to the number of correct classes in all results, while *Recall* generally refers to the recall ratio, which represents the proportion of positive categories in all the results. For object detection, if the Intersection over Union (IOU) is larger than a given threshold and positive samples are detected, this is considered a True Positive *TP*, which means that the correct sample prediction is positive; in other words, the IOU is larger than

the threshold. When the IOU of the prediction box is less than the threshold provided by the system, the correct sample prediction is marked as negative, that is, as a False Positive $FP$; a positive class predicted as a negative class is a False Negtive $FN$, while a negative class predicted correctly is a True Negative $TN$.

In order to draw the Precision–Recall (PR) curve, the detection results are first sorted by the system will sort according to the descending order of the confidence score. After assigning a value (rank), only accuracy and recall scores higher than this value are calculated. Changing the rank value changes the final recall value. We set eleven recall values ($[0, 0.1, . . . , 0.9, 1.0]$), equal to eleven rank values, and selected eleven confidence thresholds. Generally speaking, to calculate mAP it is necessary to first calculate the average precision $AP$ of each sample. Here, $AP$ is defined as the mean value of the precision after eleven recalls, with the mean value of the eleven maximum precision values taken as the mean precision. The formula is as follows:

$$AP = \frac{1}{11} \sum_{r \in 0.0.1,...,1} p_{interp}(r) \qquad where \qquad p_{interp}(r) = \max_{\tilde{r}:\tilde{r}>=r} p(\tilde{r}) \qquad (2)$$

In the formula, $p(\tilde{r})$ refers to the measurement accuracy when it is recalled to $\tilde{r}$. Here, $p_{interp}(r)$ takes only the maximum precision where the recall rate exceeds $r$. When calculating $AP$, the prediction box is first calculated and sorted. Second, a value of k is set to calculate the respective $FP$ and $TP$ values to ensure that the recall is equal to the above eleven values, then the calculation accuracy is determined. Third, the eleven accuracies are averaged to obtain the $AP$. The mAP is the sum of all $AP$ values divided by the total number of $AP$ values. FPS stands for the frame rate per second. FPS represents the real-time speed of the object detection model, and represents the total number of images that can be detected per second. In addition, the time taken to detect an image has been used by many researchers to evaluate the detection speed of object detection models.

### 3.1.1. mAP and Loss

The mAP curve transformation during SSD, FSSD, and FFSSD training (a total of 5000 training times) is shown in Figure 9.

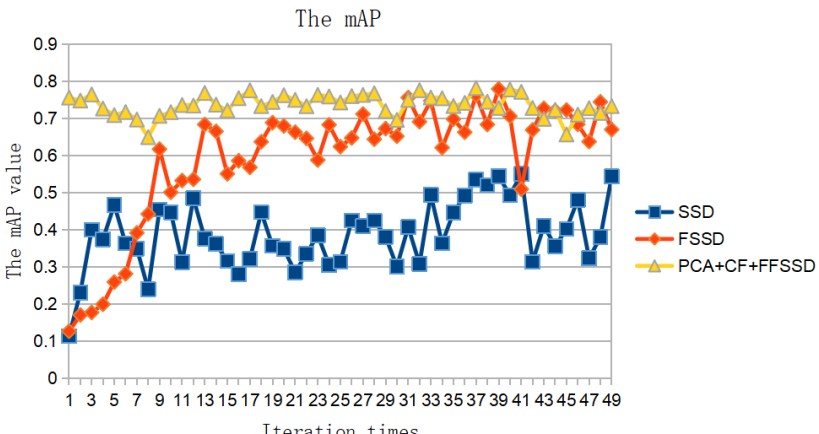

**Figure 9.** The mAP curve transformation during SSD, FSSD, and PCA+FFSSD training (a total of 5000 training instances).

Because of the participation of the two levels, namely, feature fusion and pre-training weight, the starting mAP of FFSSD is high. When the batch size is equal to 1, the mAP of the FSSD fluctuates greatly.

The transformation of the loss curve (5000 times in total) during SSD, FSSD, and FFSSD training is shown in Figure 10.

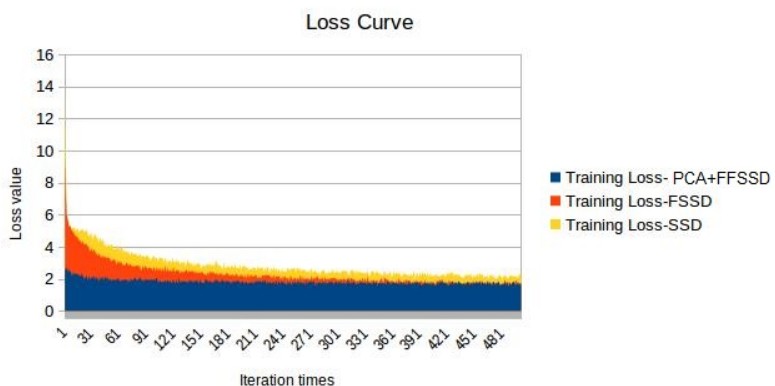

**Figure 10.** Loss curve transformation during SSD, FSSD, and FFSSD training (a total of 5000 training times).

Because of the participation of two-stage feature fusion and pre-training weight, the loss of FFSSD decreases rapidly.

### 3.1.2. Feature Visualization

A feature visualization comparison of the conv3_3, conv4_3, conv5_3, and fc7 features of the convolution layer of FSSD and FFSSD is shown in Figure 11.

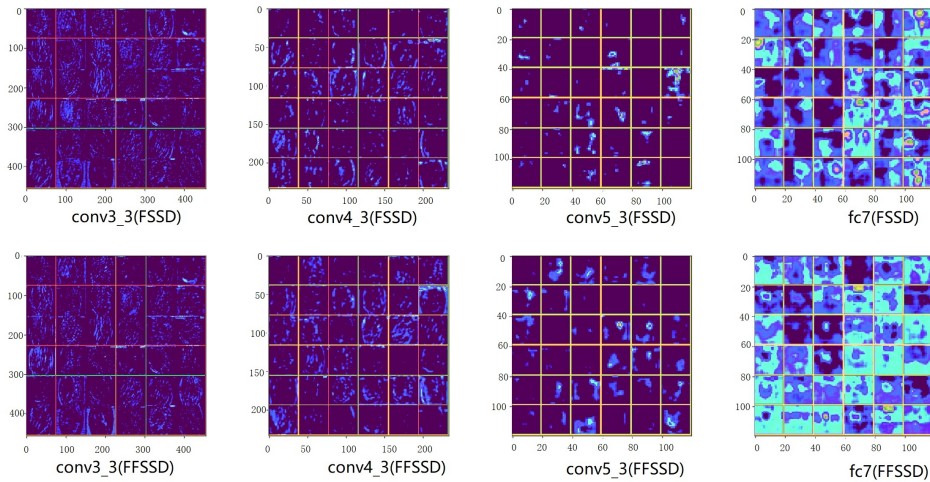

**Figure 11.** Feature visualization comparison of conv3_3, conv4_3, conv5_3, and fc7 features of the convolution layer of FSSD and FFSSD.

Compared with FSSD, the convolution layer in FFSSD can obtain more and richer features due to the addition of FPN and PCA. FSSD performs multi-fold downsampling based on the fusion of three convolution layers, then performs multi-scale detection. FFSSD performs multifold downsampling on the basis of three-times fusion of three convolution layers. In FPN, the upper convolution is fused with the lower convolution by double upsampling. The convolution layer after fusion is convolved by BatchNorm and $1 \times 1$, which makes the model converge quickly during training, eliminates the noise caused by the upper convolution, and provides more nonlinear factors.

### 3.1.3. Speed and Precision Comparison

We first show a speed and precision comparison of FSSD, FSSD, and PCA+FFSSD on NVIDIA 1070 (Table 1). The speed is expressed as img/s , that is, the time required to detect an image.

**Table 1.** Speed and Precision comparison on NVIDIA 1070.

| Model | Backbone Network | mAP | img/s |
|---|---|---|---|
| SSD | VGG16 | 71.90% | 0.027 |
| FSSD | VGG16 | 73.82% | 0.037 |
| FSSD+FPN | VGG16 | 75.45% | 0.046 |
| PCA+FFSSD | VGG16 | 82.39% | 0.046 |

The proposed FSSD includes a new feature fusion model based on SSD, and on this basis, mAP improved by 1.9%, with a 1.62% increase when FPN is used on top of FSSD. In the case of small samples, PCA was used to accelerate the convergence, pre-training weight, and multi-stage feature fusion, achieving a final the detection precision of 82.35%. In terms of time, FSSD takes 4 s to detect all test images, SSD takes 3 s to detect all test images, and PCA+FFSSD takes 5 s to detect all test images. Another very important reason why PCA+FFSSD has such a high precision is that we use the weight of FSSD+FPN training 50,000 times as the initial pre-training weight of PCA+FFSSD, meaning that the loss and mAP are in relatively low and relatively high positions, respectively. While FFSSD is nearly twice as fast as SSD, mAP has results of 82.35% in real-time. Compared with FSSD, the addition of FPN slows down FFSSD, while the detection precision of FFSSD is improved due to the top-down connection, the lateral connection between convolutional layers, the bottom-up connection, and the addition of pre-training weights. FFSSD is able to transmit information from both high-level to low-level and from low-level to high-level. The pre-training weight of FSSD+FPN makes the starting point of FFSSD very high, and the loss value is very low; thus, the detection precision is greatly improved.

Next, we provide a comparison with other object detection algorithms such as Faster R-CNN (VGG16), Faster R-CNN (ResNet 50+FPN), SSD (VGG16), SSD (ResNet-101), Efficient-D0, Efficient-D1, FSSD (VGG16), FoveaBox (ResNet+FPN), TOOD (ResNet 50+FPN), YOLOv4 (CSPDarknet-53), and Sparse R-CNN (ResNet 50+FPN) (Table 2). Their backbone networks compared algorithmically by mAP and FPS are shown in parentheses. An FPS is greater than 12.5 can be considered real-time. Faster R-CNN is only approximately real-time, and Faster R-CNN (ResNet 50+FPN) and Efficient-D7 do not reach real-time; thus, we use a short horizontal line to indicate the threshold.

**Table 2.** Comparison of various methods.

| Model | Backbone Network | mAP | FPS |
|---|---|---|---|
| Faster R-CNN [7] | VGG16 | 62.67% | 5 (K40) |
| Faster R-CNN [7] | ResNet 50+FPN | 74.41% | - |
| PCA+CF+Faster R-CNN (ours) | ResNet 50+FPN | 77.10% | - |
| SSD [5] | VGG16 | 71.90% | 46 |
| SSD [5] | ResNet-101 [5] | 74.39% | 15 (NVIDIA GTX 1070) |
| PCA+CF+SSD (ours) | VGG16 | 77.30% | 46 |
| EfficientDet-D0 [14] | B0 | 53.38% | 97 (Telsa v100) |
| EfficientDet-D1 [14] | B1 | 56.58% | 74 (Telsa v100) |
| EfficientDet-D2 [14] | B2 | 59.23% | 57 (Telsa v100) |
| EfficientDet-D3 [14] | B3 | 61.14% | 35 (Telsa v100) |
| EfficientDet-D4 [14] | B4 | 58.81% | 23 (Telsa v100) |
| EfficientDet-D5 [14] | B5 | 58.09% | 10 (Telsa v100) |
| EfficientDet-D7 [14] | B6 | 78.37% | — |
| FSSD [6] | VGG16 | 73.82% | 65.8 (NVIDIA 1080Ti) |
| PCA+CF+FSSD (ours) | VGG16 | 75.58% | 65.8 (NVIDIA 1080Ti) |
| FoveaBox [11] | ResNet50+FPN | 81.10% | 25 (NVIDIA RTX 2060) |
| TOOD [12] | ResNet50+FPN | 73.08% | 20 (NVIDIA RTX 2060) |
| PCA+CF+TOOD (ours) | ResNet50+FPN | 76.50% | 20 (NVIDIA RTX 2060) |
| YOLOv4 [10] | CSPDarknet-53 | 70.29% | 45 (NVIDIA RTX 2060) |
| PCA+CF+YOLOv4 (ours) | CSPDarknet-53 | 72.47% | 45 (NVIDIA RTX 2060) |
| Sparse R-CNN [9] | ResNet50+FPN | 75.68% | 17 (NVIDIA RTX 2060) |
| OWOD [13] | ResNet-50 | 71.30% | 62 (NVIDIA RTX 2060) |
| PCA+CF+FFSSD (ours) | VGG16 | 83.58% | 27 (NVIDIA GTX 1070) |

The proposed method has the highest detection precision and real-time detection speed. As the scale and shape of medical images can vary greatly, a general object detection algorithm is not necessarily suitable for detection of the prostate capsule, requiring verification by experiment. Image denoising can improve the mAP of algorithms such as SSD, FSSD, Faster R-CNN, YOLOv4, and TOOD. Other algorithms, such as EfficientDet, FoveaBox, Sparse R-CNN, and OWOD, do not require image denoising.

### 3.1.4. Detection Results

A comparison of the detection results of SSD, FSSD, and PCA+FFSSD is shown in Figure 12.

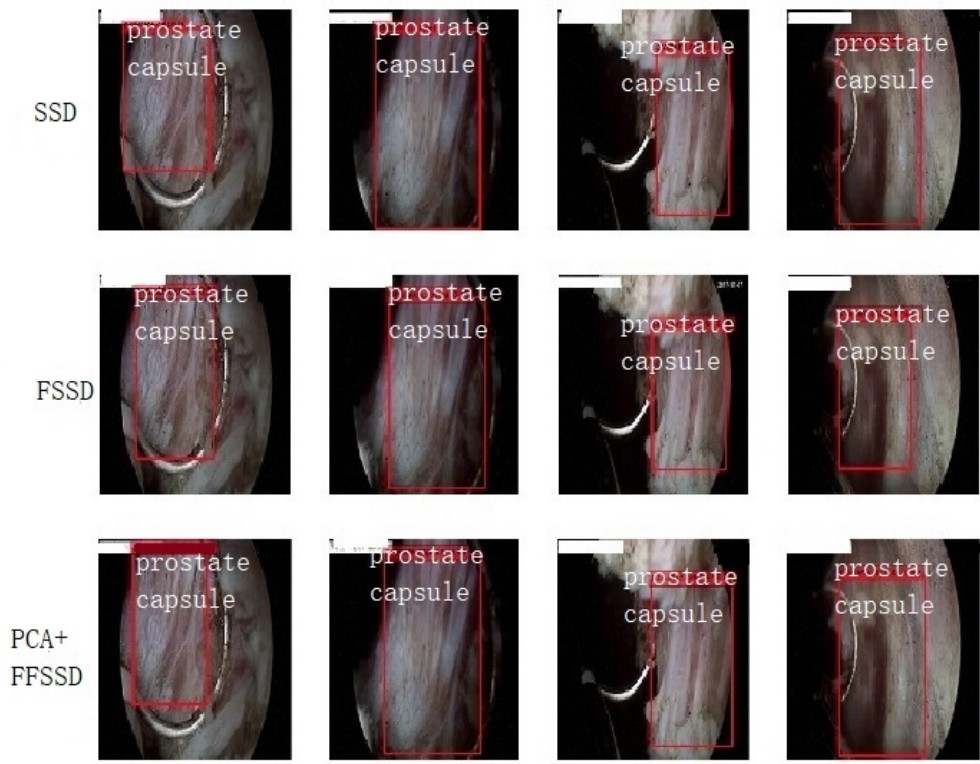

**Figure 12.** Detection result comparison for SSD, FSSD, and FFSSD.

PCA+FFSSD has better detection results and more accurate positioning. This accuracy is mainly due to the effect of blending the feature fusion mechanism of FPN and FSSD. PCA provides the effect of small sample accelerated convergence.

The mAP of FFSSD can be improved to 83.58% using PCA+FFSSD trained weights and using the PCA+CF dataset for fine-tuning during training. Because the network is unchanged, the detection speed is unchanged as well. The mAP results for ten training epochs are compared in Figure 13.

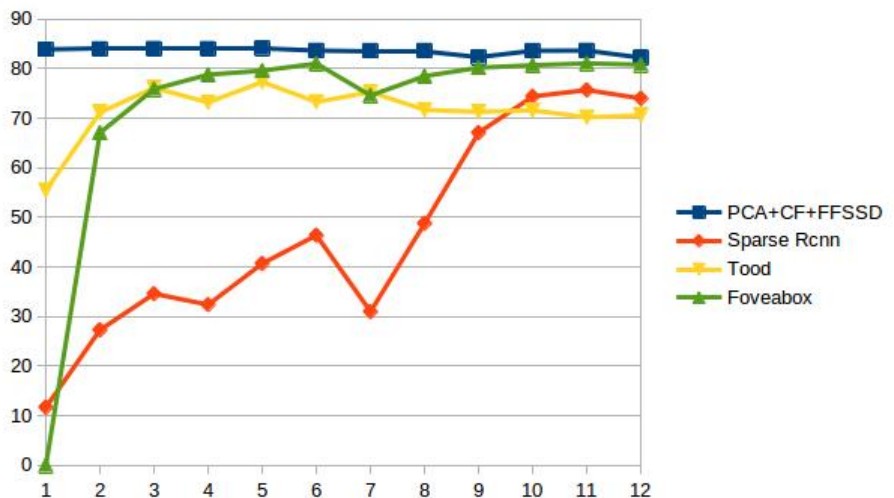

**Figure 13.** Comparison of mAP for ten training epochs.

Because of the pre-training weight, the mAP of the PCA+CF+FFSSD is relatively high; FoveaBox is second (Figure 14).

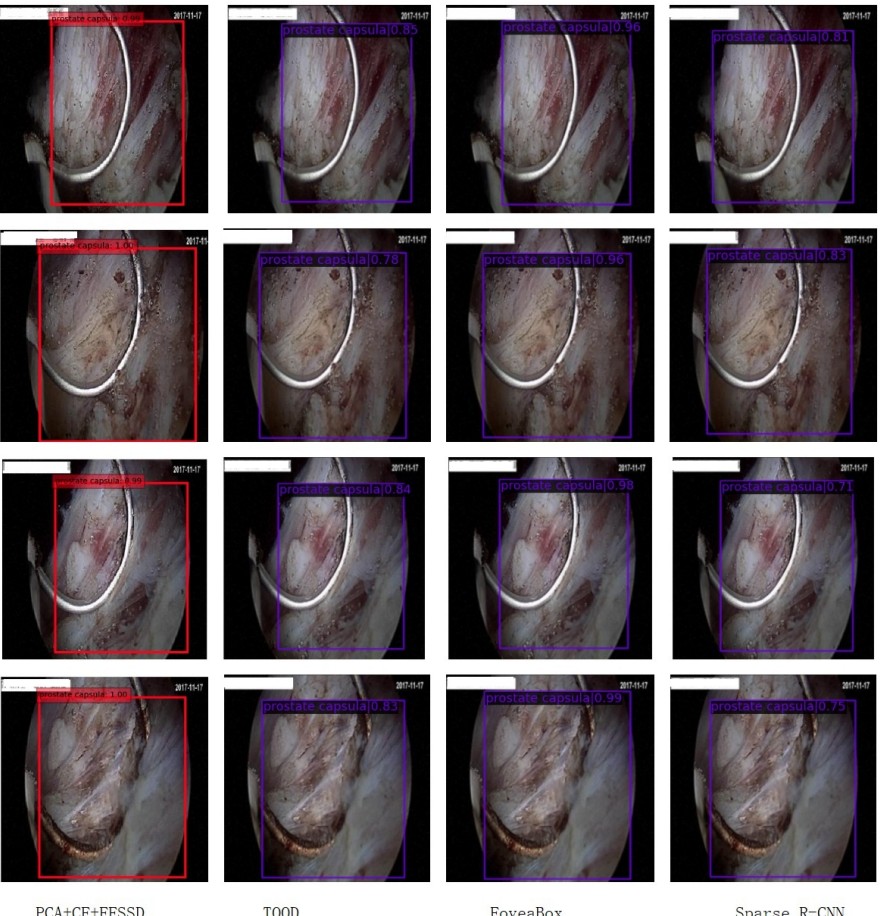

**Figure 14.** Detection result comparison for PCA+CF+FFSSD, TOOD, Sparse R-CNN, and FoveaBox.

Compared with the original model, PCA+CF+FFSSD detects the location of the prostate capsule more accurately.

## 4. Discussion and Conclusions

Although FSSD integrates the features of all three scales, it is easy to lose semantic details after multiple downsampling. However, as SSD only has one single-layer convolution to participate in subsampling, much of the location information is ignored and semantic information from the high level cannot be transferred to the low level. This paper proposes a new feature fusion mechanism that combines the feature fusion techniques of FPN and FSSD and uses top-down, bottom-up, and lateral connection methods to construct a new feature pyramid through two forward transfers and one reverse transfers. Feature visualization and ablation experiments prove the effectiveness of the proposed approach. The combination of FPN and FSSD improves the accuracy of prostate capsule detection and is able to extract more abundant features. In the later stage, the accuracy of the dataset with noise reduction is further improved. The reason for this is that during training, the pre-training weights which had been trained 50,000 times by PCA+FFSSD were used for transfer learning; thus, the accuracy of the model was further improved after fine-tuning the noise removal. In terms of detection, the multi-level network mixed method can improve the detection precision of the network. Next, we intend to try to find a method that can achieve a good balance of speed and precision. Typical object detection algorithms such as YOLOv4, Sparse R-CNN, TOOD, and FoveaBox fail to produce excellent performance in this case due to the unique deformation and multi-scale characteristics of the prostatic envelope and the sparsity of the datasets.

**Author Contributions:** Conceptualization, X.W.; methodology, S.W.; validation, S.W. and X.W.; formal analysis, S.W.; investigation, C.G.; resources, X.W.; data curation, X.W.; writing—original draft preparation, S.W.; writing—review and editing, X.W.; visualization, S.W.; supervision, C.G.; project administration, S.W.; funding acquisition, S.W. All authors have read and agreed to the published version of the manuscript.

**Funding:** This research was funded by the Scientific Research Project of Hubei Education Department, grant number B202287. Financial and technical support was provided by National Natural Science Foundation of China (grant number 2016YFC0106301)and the establishment of "5G+ Artificial intelligence" remote treatment and diagnosis platform for major aortic diseases (grant number 2022BCA035).

**Institutional Review Board Statement:** This study was approved by the Institutional Review Board Ethics Committee of Zhongnan Hospital of Wuhan University (approval number 2015029, project number 2016YFC0106301).

**Data Availability Statement:** The data presented in this study are available on request from the corresponding authors. The data are not publicly available due to the patients' privacy protection.

**Acknowledgments:** This paper was supported by the Scientific Research Project of Hubei Education Department (Detection of Medical Images Based on Feature Fusion Hierarchy Deep Network, grant number B202287).Financial and technical support was provided by National Natural Science Foundation of China (grant number 2016YFC0106301)and the establishment of "5G+ Artificial intelligence" remote treatment and diagnosis platform for major aortic diseases (grant number 2022BCA035).

**Conflicts of Interest:** The authors declare no conflict of interest.

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
