# Peer review of "Application of Feature Pyramid Network and Feature Fusion Single Shot Multibox Detector for Real-Time Prostate Capsule Detection"

_electronics, doi:10.3390/electronics12041060_

Round 1

Reviewer 1 Report

The following points need to be addressed:

1.       Typo/Grammar errors in the article throughout.

2.       The structure of the abstract is not according to the Electronics Journal. Remove headings and make them more fluent.

3.       I am afraid I have to disagree with the last sentence in the abstract rather than the converse being true.

4.       Literature is not thoroughly surveyed. References are insufficient. Add some more references like: A trainable system for object detection; Efficientdet: Scalable and efficient object detection; Foveabox: Beyond anchor-based object detection; Kalman filtering and bipartite matching based super-chained tracker model for online multi-object tracking in video sequences; Towards open world object detection; etc.

5.       Dataset portion is not acceptable.

6.       Your paper is not better than the following effort:

FSSD: Feature Fusion Single Shot Multibox Detector

7.       Compare your work with the above-mentioned article. I could not find a novice idea in the work.

Author Response

Response Letter

Thank you for your letter and for the reviewers’ comments concerning our manuscript entitled “Application of Feature Pyramid Network and Feature Fusion Single Shot Multibox Detector for Real-time Prostate Capsule Detection” (Paper ID:electronics-2077946). Those comments are all valuable and very helpful for revising and improving our paper, as well as the important guiding significance to our researches. We have studied comments carefully and have made correction which we hope meet with approval. Revised portion are marked in red in the paper. The main corrections in the paper and the responds to the reviewer’s comments are as flowing:

Q1. Typo/Grammar errors in the article throughout.

Response: This article has been embellished by MDPI and is numbered english-58296.

Q2. The structure of the abstract is not according to the Electronics Journal. Remove headings and make them more fluent.

Response: We adjusted the structure and style of the abstract. Now the abstract is as follows(we removed the headings and the MDPI english editor make them more fluent).

Abstract: During feature propagation, the low-level convolution layers of the forward feature propagation network lack semantic information, and information loss occurs when fine-grained information is transferred to higher-level convolution layers; therefore, a multi-stage feature fusion network is required to determine the interaction between low-level convolution layers and high-level convolution layers. Based on the two-way feature feedback network and the feature fusion mechanism, we created a new object detection network called the Feature Pyramid Network (FPN)-based Feature Fusion Single Shot Multibox Detector (FFSSD). A down-top and top-down architecture with lateral connections enhances the detector's ability to extract features, and high-level multi-scale semantic feature maps are utilized to generate a feature pyramid network. The results show that the mAP of the proposed network for prostate capsule image detection is 83.58$\%$, and detection can be carried out in real time. The context interaction mechanism can transfer high-level semantic information to low-level convolution layers, and convolutions produced after the fusion of low-level and high-level networks contain richer location and semantic information.

Q3.  I am afraid I have to disagree with the last sentence in the abstract rather than the converse being true.

Response:  This sentence(Down-sampling based on this can make the model learn more details and enrich the detection precision) has been removed.

Q4.  Literature is not thoroughly surveyed. References are insufficient. Add some more references like: A trainable system for object detection; Efficientdet: Scalable and efficient object detection; Foveabox: Beyond anchor-based object detection; Kalman filtering and bipartite matching based super-chained tracker model for online multi-object tracking in video sequences; Towards open world object detection; etc.

Response: Except(Kalman filtering and bipartite matching based super-chained tracker model for online multi-object tracker model for online multi-object tracking in video sequences), the paper you mentioned has all been surveyed. In addition, Task-aligned One-stage Object Detection, Sarse R-CNN, YOLOv4 was surveyed(Page 1 and Page 14).

Early object detection techniques did not use feature fusion technology[1-3]. Since the emergence of deep learning, more and more feature fusion techniques have been used in object detection networks, including the image pyramid, feature fusion for the R-CNN series, the Single Shot Multibox Detector (SSD), the Feature Pyramid Network (FPN) in the FoveaBox, Sparse R-CNN and Task-aligned One-stage Object Detection (TOOD), the Feature Fusion Single Shot Multibox Detector (FSSD), the Weighted Bidirectional feature pyramid Network (BiFPN) in You Only Look Once Version 4 (YOLOv4) and EfficientDet, the Path Aggregation Network (PANet) in YOLOv4, etc.[4-14](Figure 1).

Q5.  Dataset portion is not acceptable.

Response: The proposed network was verified on three datasets, all of which contained 597 images, 478 training datasets, and 119 test datasets(It matches the ratio of eight to two). Because the data set was adjusted, all the experiments were redone and the results were slightly different from the last time.

Q6.  Your paper is not better than the following effort: FSSD: Feature Fusion Single Shot Multibox Detector.

Response: FSSD is an excellent object detection network, as is FoveaBox, OWOD, Sparse R-CNN, TOOD, YOLOv4. But the medical image is different from the ordinary image, the deformation is serious, the scale changes greatly. It may not be appropriate to directly apply the existing algorithm to the target detection algorithm. We displayed the prostate capsule images in Figure 2 as follows(Figure 4 in page 5).

We added Figure 3(Figure 8 in paper, page 7) to illustrate the difference between FSSD and FFSSD.

The three convolution layers are fused by the FPN feature pyramid to form a new convolution layer. On this basis, the second feature forward propagation step is carried out to generate a new feature pyramid network. In Figure 8, the orange dotted line box represents the FPN feature fusion network. The dotted box represents the feature fusion network of the FSSD. The small, black, dotted-line box represents the fusion process conducted on the lower-layer feature graphs. This involves 1 transverse connection (1×1 conv) and 2x upsampling steps on the higher convolution layers. BatchNorm accelerates the convergence process of the convolutional network, and the 1×1 conv step eliminates the alialation effect caused by upsampling. The fusion details are shown in the large, black, dotted box.

In addition, we show the adaptability and superiority of the proposed algorithm from the comparison of feature mAP visualization, loss, map and detection results(Figure 4, Figure 5, Figure 6, Figure 7 in response, Figure 12, Figure 10, Figure 11, Figure 13 in paper, pages 9-12).

With the help of pre-training weight, the loss drops faster and the starting mAP of the proposed algorithm is higher. The detection results were better.

Q7. Compare your work with the above-mentioned article. I could not find a novice idea in the work.

Response: We compare our work with Foveabox, OWOD,Efficientdet(b0-b7), Sparse R-CNN,YOLOv4, .etc(Figure 7, Figure 8, Figure 9 in respone, Figure 13, 14 and table 2 in paper, pages 12-13).

As you can see from Figure 9, the algorithms differ greatly. For medical images with large deformation scale transformation, different algorithms have different detection effects. The small data set is also a reason for the different algorithm detection results.

This innovations presented in this article include the following:

(1) In order to reduce the impact of salt and pepper noise on object detection network, a salt and pepper noise reduction method based on edge feature preservation is proposed. Compared with other 14 methods, this method has the highest peak signal-to-noise ratio.

(2) Based on FPN and FSSD, a multistage feature fusion network is proposed. Compared with the other algorithms, the proposed algorithm has the highest mAP for prostate capsule detection.

Reviewer 2 Report

Comments to the Author

This manuscript proposes a feature fusion network for prostate capsule detection. Evaluation on a dataset with 597 images show that by transferring high-level semantic information to low-level convolution and fusing low-level & high-level features, richer location and semantic information could be extracted. Down-sampling based on this can make the model learn more details and enrich the detection precision.

Major comments:

1. In ‘Chapter 0. Introduction’, related works on the specific task: “prostate capsule detection” are not mentioned.

2. The captions of Figure 2 and Figure 3 are confusing, which are different from the description in the manuscript. Which are the results of CF and which are the original images? What do the red rectangle areas mean in those figures?

3. Arrows missing on the upper left side of Figure 7? (The first feature forward propagation)

4. It’s not clear why different training times are used to evaluate mAP and loss in Chapter 2.1.1.

5. It’s not described what do the two red rectangle areas mean in each subfigure of Figure 13.

6. Regarding the ‘FPS’ performance in Table 2, what’s the threshold above which the method could be determined as ‘real-time’?

Minor comments:

1. Please double check the typos, e.g.,

Line 10: prostate capsula -> capsule

Line 67: duplicate ‘as’

Line 269: prostate capuse -> capsule

2. Please double check the manuscript to avoid the grammar errors, e.g.,

Line 90-91: we propose a new multi-level feature fusion feature pyramid 90 network, a new network called FFSSD was proposed.

Author Response

Response Letter

Thank you for your letter and for the reviewers’ comments concerning our manuscript entitled “Application of Feature Pyramid Network and Feature Fusion Single Shot Multibox Detector for Real-time Prostate Capsule Detection” (Paper ID:electronics-2077946). Those comments are all valuable and very helpful for revising and improving our paper, as well as the important guiding significance to our researches. We have studied comments carefully and have made correction which we hope meet with approval. Revised portion are marked in red in the paper. The main corrections in the paper and the responds to the reviewer’s comments are as flowing:

Major comments:

Q1. In ‘Chapter 0. Introduction’, related works on the specific task: “prostate capsule detection” are not mentioned.

Response: Related works was montioned in page 2. Previous research on the prostate capsule involved the surgical level, contour recognition, thickness calculation, and volume calculation. At the surgical level, Zia et al. explored the U-shaped incision of the prostate capsule in laparoscopic prostatectomy, and Alon Z Weizer evaluated the risk of developing urothelial and prostate cancers in a series of patients undergoing radical cystoprostatectomy with a preserved prostate capsule [19,20]. Rania Hussein used prostate shape and the generalized Hough transform to automatically recognize the contours of the prostate capsule [21]. KG Holder et al. studied the relationships among the prostate volume, prostate capsule thickness, and gland epithelial cell density in the peripheral zone (PZ) by using tissue anatomical imaging technology [22].

Q2. The captions of Figure 2 and Figure 3 are confusing, which are different from the description in the manuscript. Which are the results of CF and which are the original images? What do the red rectangle areas mean in those figures?

Response: We change the caption of figure 2(Figure 1 in paper, page 3) and figure 3(Figure 2 in paper, page 4). The origin means the origin image. The CF compose 6 algorithms(AMF, NAMF, ANN, AMF-ADF-ANN, NAMF-ADF-ANN, AMF-ADF-ANN, CF take the maximum based on psnr). NAMF-ADF-ANN means CF algorithm here(We changed NAMF-ADF-ANN to CF, Figure 2 in paper, page 4). The red rectangle areas here remind people to pay attention to the details after noise reduction.

Q3. Arrows missing on the upper left side of Figure 7? (The first feature forward propagation)

Response. We have added the upper left side in Figure.(Now Figure 9 in paper, page 9.)

Q4. It’s not clear why different training times are used to evaluate mAP and loss in Chapter 2.1.1.

Response. We have uniformly adjusted the number of training times to 5000.

Because of the participation of two levels of feature fusion and the pretraining weight, the starting mAP of the FFSSD was high. When the batchsize was equal to 1, the mAP of the FSSD fluctuated greatly. The transformation of the loss curve (5000 iterations in total) during SSD, FSSD, and FFSSD training is shown in Figure 11.

Q5. It’s not described what do the two red rectangle areas mean in each subfigure of Figure 13(Now Figure 15 in page 13).

Response. Due to the proportion of data sets, we did the experiment again and chose a larger nms threshhold(0.5). In addition, some experimental contrasts (TOOD, Sparse R-CNN, YOLOv4, etc.) were added. Now the comparisons were displayed in Figure 14 and Figure 15.

Compared with the original model, the PCA+CF+FFSSD models was shown to detect the location of the prostate capsule more accurately.

Q6. Regarding the ‘FPS’ performance in Table 2, what’s the threshold above which the method could be determined as ‘real-time’?

 Response. The parentheses show the backbone networks of the algorithms, which were compared algorithmically by mAP and FPS. When the FPS is greater than 12.5, it can be considered to represent real-time(Page 11,Line 286-287).

Minor comments:

  1. Please double check the typos, e.g.,

Line 10: prostate capsula -> capsule

Line 67: duplicate ‘as’

Line 269: prostate capuse -> capsule

Response. This paper is edited by MDPI(english-28596.)

  1. Please double check the manuscript to avoid the grammar errors, e.g.,

Line 90-91: we propose a new multi-level feature fusion feature pyramid 90 network, a new network called FFSSD was proposed.

Response. Based on the FPN and FSSD, we propose a new multi-level feature fusion feature pyramid network, FFSSD(Line 125-126, page 4).

Round 2

Reviewer 1 Report

I could not find a novice idea in the work.

Author Response

Response Letter(Round 2)

Thank you for your letter and for the reviewers’ comments concerning our manuscript entitled “Application of Feature Pyramid Network and Feature Fusion Single Shot Multibox Detector for Real-time Prostate Capsule Detection” (Paper ID:electronics-2077946). Those comments are all valuable and very helpful for revising and improving our paper, as well as the important guiding significance to our researches. We have studied comments carefully and have made correction which we hope meet with approval. Revised portion are marked in red in the paper. The main corrections in the paper and the responds to the reviewer’s comments are as flowing:

Q1. I could not find a novice idea in the work.

Response: The article’s innovations include:

(1) In order to reduce the impact of salt and pepper noise on object detection network, a salt and pepper noise reduction method based on edge feature preservation is proposed. Compared with other 14 methods, this method has the highest peak signal-to-noise ratio. The proposed image-denoising methods make the mAP of Faster R-CNN, YOLOv4, TOOD, SSD and FSSD higher.

(2) Based on FPN and FSSD, a multistage bidirectional feature fusion network FFSSD is proposed. Compared with other algorithms(such as Faster R-CNN, YOLOv4, TOOD, SSD, FSSD, OWOD, Foveabox, Sparse R-CNN, Efficientdet and so on), the proposed algorithm has the highest mAP of prostate capsule detection(Page 2, line 79-88)..

Thank you for your valuable comments in the review. The quality of the revised articles has been greatly improved. If you have any other comments, please let me know in time.

Best regards,

Shixiao Wu

Reviewer 2 Report

Comments to the Author

This manuscript proposes a feature fusion network for prostate capsule detection. Evaluation on a dataset with 597 images show that by transferring high-level semantic information to low-level convolution and fusing low-level & high-level features, richer location and semantic information could be extracted. Down-sampling based on this can make the model learn more details and enrich the detection precision.

Minor comments:

1. Seems the sample images in Figure 4 are in different angles. It’s better to normalize the picture to the same angle, unless there is a specific purpose in doing so.

2. It’s not easy to read the three different bars in Figure 10, especially the red and yellow ones. It’s better to modify the graph to show them more clearly.

Author Response

Thank you for your letter and for the reviewers’ comments concerning our manuscript entitled “Application of Feature Pyramid Network and Feature Fusion Single Shot Multibox Detector for Real-time Prostate Capsule Detection” (Paper ID:electronics-2077946). Those comments are all valuable and very helpful for revising and improving our paper, as well as the important guiding significance to our researches. We have studied comments carefully and have made correction which we hope meet with approval. Revised portion are marked in red in the paper. The main corrections in the paper and the responds to the reviewer’s comments are as flowing:

Q1. Seems the sample images in Figure 4 are in different angles. It’s better to normalize the picture to the same angle, unless there is a specific purpose in doing so.

Response: Figure 4 is rewitten(Page 5).

Q2. It’s not easy to read the three different bars in Figure 10,  especially the red and yellow ones. It’s better to modify the graph to show them more clearly.

Response: Figure 10 is rewitten(Page 9).
